# Impact of Hydrogen Peroxide on Protein Synthesis in Yeast

**DOI:** 10.3390/antiox10060952

**Published:** 2021-06-12

**Authors:** Cecilia Picazo, Mikael Molin

**Affiliations:** 1Institute for Integrative Systems Biology, I2SysBio, University of Valencia-CSIC, 7, 46980 Paterna, Spain; 2Department of Biology and Biological Engineering, Chalmers University of Technology, SE-412 96 Gothenburg, Sweden

**Keywords:** hydrogen peroxide, cysteine thiols, protein synthesis, signaling

## Abstract

Cells must be able to respond and adapt to different stress conditions to maintain normal function. A common response to stress is the global inhibition of protein synthesis. Protein synthesis is an expensive process consuming much of the cell’s energy. Consequently, it must be tightly regulated to conserve resources. One of these stress conditions is oxidative stress, resulting from the accumulation of reactive oxygen species (ROS) mainly produced by the mitochondria but also by other intracellular sources. Cells utilize a variety of antioxidant systems to protect against ROS, directing signaling and adaptation responses at lower levels and/or detoxification as levels increase to preclude the accumulation of damage. In this review, we focus on the role of hydrogen peroxide, H_2_O_2_, as a signaling molecule regulating protein synthesis at different levels, including transcription and various parts of the translation process, e.g., initiation, elongation, termination and ribosome recycling.

## 1. Introduction

All eukaryotic organisms utilize oxygen-dependent metabolism because, through evolution, this molecule has been selected as a final electron acceptor in respiration in most cases. The consequence of that is that all aerobic organisms are under the paradox of oxygen [1]. While aerobic organisms depend on oxygen in cellular respiration, its oxidant power produces cytotoxic compounds called reactive oxygen species (ROS), several of which are unstable oxygen species containing unpaired electrons, so-called free radicals. ROS form in vivo primarily in oxidative phosphorylation during mitochondrial respiration, and when their levels overwhelm cellular defenses, oxidative stress causes widespread damage to most macromolecules (lipids, proteins and DNA). Although the name ROS refers to the reactive nature of the atoms, ROS encompasses a group of molecules that are vastly different, spanning up to 11 orders of magnitude in their respective second-order rate constants with specific targets. Hence, speaking of ROS as a group leads to chemical imprecision [2]. In human cells, a total of 41 hydrogen peroxide (H_2_O_2_)- and/or superoxide anion (O_2_^•−^)-generating enzymes have been found [3]. Representative intracellular sources of ROS production are nicotinamide adenine dinucleotide phosphate (NADPH) oxidase, xanthine oxidase, endoplasmic reticulum (ER) protein folding, peroxisomal fatty acid metabolism and mitochondria. Among them, the electron transport chain of the mitochondria is usually considered to be the major source, producing over 80–90% of ROS [4]. However, under certain conditions ER oxidative folding has been estimated to account for ~25% of all ROS produced [5]. During respiration, electrons go to molecular oxygen (O_2_) that is reduced to water. However, a continued influx of electrons also leads to increases in various types of ROS.

The most well-known ROS produced in this way is the superoxide anion radical, O_2_^•−^, which is converted into hydrogen peroxide (H_2_O_2_) through dismutation. O_2_^•−^ itself may oxidize Fe-S clusters, whereas H_2_O_2_ reacts mainly with the sulfur-containing amino acids cysteine and methionine. On the other hand, hydroxyl radical, ^•^OH, that can be produced in reactions between H_2_O_2_ and metal ions (via the Fenton reaction), is considered to be the most powerful oxidant because it can react with almost any biomolecule. Protein oxidation by the least reactive of the ROS typically occurs at the level of specific amino acid residues. For example, H_2_O_2_ oxidizes the thiol group (-SH) of a cysteine (Cys) residue to form a sulfenic (-SOH) that may, e.g., result in disulfide bond formation (-S-S-). Excess H_2_O_2_ can generate cysteine sulfinic (-SO_2_H) and sulfonic (-SO_3_H) acid forms. ROS can also oxidize the sulfur atom of methionine, leading to its conversion into methionine sulfoxide or methionine sulfone [6]. The major impact of H_2_O_2_ on redox regulation occurs via thiol peroxidases and involves reversible protein cysteine oxidation to the sulfenylated or disulfide forms. The median percentage of oxidation of cysteine residues in the proteome is between 5–12% and this can be increased up to 40% by oxidants [7]. However, due to their instability and the relative lack of methods to study them, the scope of redox regulation via cysteine sulfenylation is still poorly understood.

H_2_O_2_ is the most stable of the ROS and uncharged. It is (freely) diffusible and reactive but must be removed from cells to avoid Fenton and Haber–Weiss reactions leading to the formation of highly reactive hydroxyl radicals. H_2_O_2_ emerged as the major redox metabolite operative in redox sensing, signaling and redox regulation and is the main and most abundant primary reactive oxygen species formed in mammalian cells, either as a side product or even intentionally by enzymes [8].

As a messenger molecule, H_2_O_2_ diffuses through cells and tissues to initiate immediate cellular effects, e.g., initiation of proliferation and recruitment of immune cells to tumors [9]. This means that H_2_O_2_ can carry a redox signal from the site of its generation to a target site. At low concentrations, ROS can act as a second messenger involved in signaling pathways and produce beneficial effects (e.g., support health and longevity), a phenomenon called mitohormesis, as originally observed using moderate levels of the redox-cycling drug paraquat, acting in mitochondria to increase the levels of ROS [10]. In their recent review, Sies et al. described two new concepts, oxidative eustress for the beneficial effects of H_2_O_2_ and O_2_^•−^ and oxidative distress for the irreversible damage that can result from higher levels of H_2_O_2_ and O_2_^•−^ [3,9].

As an example of ROS acting as a signaling molecule, we have shown that light is converted by a peroxisomal oxidase into an H_2_O_2_ signal that is sensed by the yeast peroxiredoxin Tsa1 to inhibit protein kinase A (PKA)-dependent phosphorylation of the yeast general stress-responsive Zn finger transcription factor Msn2 [11]. Furthermore, it has recently been shown that in the yeast metabolic cycle (YMC), a cyclic metabolic adaptation to stress that has been proposed to model adaptations to the prevailing recurring alterations of sunlight and darkness (circadian rhythms) occurring in many organisms, DNA replication and, thus, the cell cycle are restricted to a reducing phase, characterized by low levels of oxidants and that this is modulated through H_2_O_2_-dependent protein thiol oxidation and the cytosolic peroxiredoxins Tsa1 and Tsa2 [12].

Imbalances between ROS and the antioxidant defense mechanisms are thought to play an important role in carcinogenesis and in responses to chemotherapy, and are linked also to age-related diseases such as Parkinson’s. To counteract oxidative stress, cells have protection systems and ROS detoxification systems to prevent damage at the macromolecular level.

This review describes the role of H_2_O_2_ in the different stages of protein synthesis, from transcription to translation, focusing on the role of H_2_O_2_ as a signaling molecule.

## 2. Systems Involved in Responses to H_2_O_2_

There are overall two different main redox systems relevant to and involved in H_2_O_2_ homeostasis, one being the cysteine–thiol redox system (composed mainly of the thioredoxin and glutathione/glutaredoxin systems), and the other one, the H_2_O_2_ detoxification system. The thioredoxin and the glutaredoxin systems are involved mainly in redox balance. Thioredoxins (TRXs) are oxidoreductases with two cysteines in their active sites that are implicated in thiol reduction. Their most important function is thought to be in making possible the catalytic cycles of the interacting peroxidases. Glutaredoxins (GRXs) (in the glutaredoxin system) are responsible for the reduction of protein disulfides or glutathione–protein mixed disulfides. The GRX system also includes NADPH and glutathione reductase. GRXs carry cysteines in their active site and use thioredoxin or glutathione as hydrogen donors [13,14].

In the system of cellular detoxification, the most abundant non-enzymatic antioxidant molecule is glutathione (GSH) [15], whereas other molecules, such as ascorbic acid, are thought to be important too. In mammalian cells, and other organisms capable of synthesizing the selenocysteine amino acid, GSH directly reduces the highly efficient and selenocysteine-dependent glutathione peroxidase enzymes [16]. However, the role of GSH in H_2_O_2_ scavenging in yeast, *Saccharomyces cerevisiae*, that cannot synthesize selenocysteine, is unclear, since it was shown that neither cells depleted of or containing elevated, toxic levels of GSH, were affected in thiol redox maintenance except for higher levels of GSH blocking oxidative protein folding and secretion [17,18]. Interestingly, the most important function of GSH in yeast was instead proposed to be linked to iron metabolism because of an essential requirement in Fe-S cluster biogenesis.

Key components in the detoxification system are enzymes that act as ROS detoxifiers. The antioxidant enzymes are superoxide dismutase (SOD), catalase (CAT), glutathione peroxidase (GPx), which employs GSH, and thioredoxin peroxidase (TPx) (most of which are of the peroxiredoxin family), which employs TRXs as reductants. It is worth noting, however, that several of the yeast glutathione peroxidase sequence-similar enzymes are functional TPxs that are re-reduced by thioredoxin (e.g., Gpx3, [19]), consistent with the redundancy of glutathione in cytosolic thiol redox maintenance in this species. Enzymes in the peroxiredoxin family (Prx) do not use a cofactor like the other antioxidant enzymes; they use one or two cysteines (Cys) in the catalytic cycle (either only the peroxidatic cysteine, C_P_, in the case of 1-Cys Prx, or C_P_ and the resolving cysteine, C_R_, in the cases of 2-Cys Prxs) to mediate the reduction of intracellular H_2_O_2_, and the thioredoxin/thioredoxin reductase system as an electron donor to support catalysis. 

Peroxiredoxins use reactive cysteine amino acids to decompose hydrogen peroxide, to regulate cellular signaling by transducing hydrogen peroxide signals and to bind to misfolded proteins (via their chaperone function [20,21]). In yeast, there are five peroxiredoxins (Prx), three of which are cytoplasmic (Tsa1, Tsa2 and Ahp1), one nuclear (Dot5) and one 1-Cys peroxiredoxin which functions in the mitochondria, Prx1 [22]. Mammalian cells express six peroxiredoxins, out of which four are 2-Cys peroxiredoxins (Prx1, Prx2, Prx3 and Prx4), one an atypical 2-Cys Prx (Prx5) and one a 1-Cys Prx (Prx6) [23].

Low levels of H_2_O_2_ cause the oxidation of Prx into a sulfenic form (Cys-SOH) which may form an intermolecular disulfide bond with another subunit of the Prx dimer. The resulting disulfide bond may be reduced back via the thioredoxin/thioredoxin reductase system for an effective removal of intracellular ROS. High levels of H_2_O_2_ can cause hyperoxidation into a sulfinic form (Cys-SO_2_H) that stabilizes a high-molecular-weight (HMW) complex and inactivation of the peroxidase function. The HMW form has been associated with a functional switch of the enzyme into a chaperone holdase activity, that seems to be modulated also by its dissociation via sulfiredoxin, Srx1 [21]. Persistently elevated H_2_O_2_ levels transform Prx catalytic cysteines into an irreversible sulfonic acid form (Cys-SO_3_H), also multimerizing into a HMW form. 

Because Prx peroxidase activity is inactivated by hyperoxidation, Karplus et al. proposed the ‘flood-gate’ model of H_2_O_2_ signaling [24]. In this model, Prx inactivation allows H_2_O_2_ to accumulate to levels inhibiting other signaling enzymes (e.g., protein tyrosine phosphatases) [25]. However, Prx inactivation typically occurs at levels of H_2_O_2_ higher than needed for signaling and life-span extension and is independent of the resolving cysteine, a key determinant of Prx anti-aging [26,27]. Thus, other mechanisms must be involved in Prx-regulated mitohormesis. Stöcker S et al. noted that the affinities for H_2_O_2_ of peroxiredoxins are many orders of magnitude higher than that of other cellular systems. In fact, it has remained unclear if other proteins involved in signaling could become oxidized directly by H_2_O_2_. In fact, the authors noted that deletion of cytosolic Prxs suppressed overall H_2_O_2_-dependent protein thiol oxidation (disulfide bonds), suggesting that Prxs are highly efficient enablers of protein thiol oxidation and redox signaling [28].

Peroxiredoxins may also participate in different cellular processes, such as cell growth, apoptosis, neuronal differentiation and cancer through their interaction with other proteins [29,30]. 

In yeast, the most highly expressed of the peroxiredoxins is the cytosolic enzyme Tsa1. We have recently shown that Tsa1 plays key roles in both age-related proteostasis and in kinase signaling. In particular, in cells lacking Tsa1, chaperones fail to recognize and bind age-related damaged and aggregated proteins, suggesting a key role of the Tsa1 chaperone function in age-related protein quality control [31]. On the other hand, cells lacking Tsa1 also display aberrant protein kinase signaling, directly impinging on both H_2_O_2_ defenses and the rate of aging [27].

## 3. Redox Regulation of Protein Synthesis

Cells must be able to maintain intracellular homeostasis in the face of changing conditions. A common response to some stress conditions (as oxidative stress) is the global inhibition of protein synthesis, of which the inhibition of translational initiation is the best understood mechanism of stress-induced translation control [32,33]. The process of protein synthesis is expensive, consuming much of the cell’s energy. Consequently, it must be regulated to conserve resources. Because of this, the regulation of protein synthesis is also intimately connected to nutrient signaling, e.g., via the target of rapamycin (TOR) and PKA pathways.

The TOR pathways contain members of the phosphatidylinositol-3-kinase related kinase family assembling into two different multisubunit complexes that are conserved from yeast to mammals, TORC1 and TORC2. TORC1 senses metabolic stress by assessing cellular nutrient and energy status, whereas TORC2 regulates cytoskeleton and cellular proliferation by sensing different growth factors. Upon high nutrient availability and non-stress conditions the TORC1 pathway is active, promoting gene transcription, transcription of ribosomal proteins (RP) and ribosomal biogenesis and repressing the transcription of starvation-specific genes, autophagy and as a final consequence, growth of the cells [34,35,36].

Equally importantly, cells need to make proteins in the right amount and at the right time and place. Therefore, translational control is an important contributor to regulating gene expression and many disease states occur because of its dysfunction [37]. A reduction in protein biosynthesis may prevent continued gene expression during stress conditions as well as allow the turnover of existing mRNAs and proteins such that gene expression can be reprogrammed as a response to stress. Translation rates are not uniform among different mRNAs and are commonly determined by the proteins bound to them to form messenger ribonucleoprotein complexes (mRNPs) [38]. Specific protein factors within mRNPs dictate the translational status (stimulate or inhibit translation) and their localization into RNA granules [39]. Another factor determining translational rates is the availability of transfer RNAs (tRNAs) and the codon composition of an mRNA [40], as we will comment on later in this review.

Topf et al. in 2018 found via a global investigation of cysteine residues oxidized by H_2_O_2_ at a previously unrivalled depth that exposing yeast to a moderate level of H_2_O_2_ targets specific proteins rather than indiscriminately oxidizing cysteine thiols [41]. Notably, despite that the overall change in protein thiol oxidation was not greater than plus 13% on average, the authors identified redox-sensitive cysteine residues in 47 proteins that are likely regulated in a reversible way by H_2_O_2_. Mapping of redox-sensitive thiols upon exposure to endogenous mitochondrially produced or exogenous H_2_O_2_ revealed a striking overrepresentation of protein thiols in the protein translation apparatus. Thus, thiol oxidation appears to preferentially target protein synthesis. For that reason, this review focuses on the important role of H_2_O_2_ in protein synthesis at different levels.

The process of protein synthesis can be divided into transcription and translation parts. Whereas the regulation of protein synthesis through transcription is overall significantly more time consuming, it may play a more prominent role in adaptation [26]. On the other hand, translational control allows more immediate control over the protein synthesis process [42].

## 4. Role of H_2_O_2_ in Protein Synthesis: Transcription

There are different mechanisms by which H_2_O_2_ modulates transcription. From bacteria to mammals, one way that this is operated is through the regulation of the activity of transcription factors (TF) [43]. In bacteria that do not have cellular compartments, transcription factors are direct sensors of H_2_O_2_ to enable a fast response. There are two TFs in the bacterium *Escherichia coli*, which are highly reactive to H_2_O_2_, OxyR and PerR. In yeast, a eukaryotic but still unicellular organism, there are five different TFs that are activated in response to ROS: Yap1, Skn7, Maf1, Hsf1 and Msn2/4 (Figure 1). In multicellular organisms such as mammals, H_2_O_2_ evolved as a second messenger necessary for many signaling pathways. Here, nine TFs can be classified as relevant in the response to ROS, namely AP-1, NRF2, CREB, HSF1, HIF-1, TP53, NF-κB, NOTCH, SP1 and SCREB-1. NRF2, together with its inhibitor, Keap1, may be the most important in the response against oxidative stress. Under oxidative stress conditions, Nrf2 detaches from Keap1 and is transported to the nucleus where it (in a heterodimer form with Maf proteins) recognizes the antioxidant response element (ARE) that is essential for transcription of antioxidant and metabolic genes [44].

Yeast Yap1, a basic leucine zipper (bZIP) transcription factor, is regulated by H_2_O_2_ at the level of its transport between the nucleus and the cytoplasm. Under oxidative stress, nuclear export of Yap1 is inhibited and Yap1 is retained in the nucleus, where it stimulates the expression of target genes [45]. Cytosolic thioredoxin 2, Trx2, and glutathione peroxidase 3, Gpx3, are involved in the Yap1 H_2_O_2_ response as reducing agents and redox relay receptors, respectively. Yap1 has six cysteines, four of which form disulfide bonds (C303-C598 and C301-C629) upon H_2_O_2_ addition. In Yap1 activation, Gpx3 acts as the H_2_O_2_ sensor, and following Gpx3 oxidation by H_2_O_2_, a disulfide bond between Gpx3-Cys36 and Cys598 in Yap1 forms, which is resolved into a Yap1 intramolecular disulfide bond, causing Yap1 to accumulate in the nucleus [19].

Both the Yap1 and Skn7 transcription factors participate in the response to H_2_O_2_. The connection between Yap1 and Skn7 upon H_2_O_2_ promotes the expression of the *TRX2* and *TRR1* (cytosolic thioredoxin reductase) genes. Indeed, Yap1 controls the expression of different genes under H_2_O_2_ conditions, most of which require the presence of both Yap1 and Skn7. Some of these genes include genes important in the regulation of ROS signaling, such as the two superoxide dismutases (SODs), the cytosolic copper–zinc SOD, Sod1, and the mitochondrial manganese SOD, Sod2; the dominantly expressed cytosolic 2-Cys peroxiredoxins, *TSA1* and *AHP1*; and other genes, such as *SSA1* (encoding a major Hsp70 protein) [46]. Anthony G Beckhouse et al. showed in 2008 that adaptation to H_2_O_2_ treatment occurs when anaerobically grown wild-type cells are aerated for a short time and depends on the Yap1 and Skn7 transcription factors [47], indicating that transcriptional responses to H_2_O_2_ are activated also in response to oxygenation. 

Heat shock transcription factor 1, Hsf1, regulates the transcription of genes that contain a pentameric heat shock element (HSE), repeating units of nGAAn, under normal conditions. In fact, *HSF1* controls the transcription of numerous genes encoding proteins other than heat shock proteins (HSPs), largely in a temperature-independent manner, so-called *non-HSP target genes* of *HSF1*. These include genes related to longevity, immune response, autophagy and oxidant defense (more information in [48]). In agreement with a close connection between responses to heat shock and oxidative stress, cells lacking the antioxidant enzymes catalase or SODs are hypersensitive to heat shock [49], and Sugiyama et al. showed in 2000 that *SOD2* is required for resistance to heat-induced oxidative stress [50]. In yeast exposed to H_2_O_2_, Hsf1 binds to the promoters of *CUP1*, *BTN2*, *SIS1*, *ERO1*, *SGT2* and *SSA3* genes and upregulates their transcriptional activity [51], similar to mammalian *HSF1* which responds to the treatment of H_2_O_2_. In mammalian cells, *HSF1* has two cysteines (Cys 35 and Cys 105) in the DNA-binding domain that form reversible disulfide bonds under H_2_O_2_ conditions, which causes binding of Hsf1 to the HSE and activation of HSP gene expression [52]. In *S. cerevisiae*, Hsf1 completely lacks cysteine residues in its DNA-binding domain, indicating that Hsf1 is not a direct target of thiol oxidation. However, it has been noted that cysteines in the Hsf1-interacting and Hsf1-inhibiting yeast cytosolic Hsp70 Ssa1 [53] are absolutely required for the Hsf1 response to thiol-reactive compounds [54]. Nevertheless, in *Drosophila* and mammals there is a conformational change in Hsf1 upon oxidative stress and *HSF1* is necessary to maintain redox homeostasis and antioxidant defenses [51].

Skn7 shares certain structural homologies with Hsf1, particularly in its DNA-binding domain. Under t-butyl hydrogen peroxide (t-BOOH) treatment, Hsf1 interacts physically and genetically with Skn7 to achieve maximal induction of several HSP genes [55].

Msn2 and Msn4 (Msn2/4) are homologous and functionally redundant Cys_2_His_2_ zinc finger yeast TFs. Disruption of both results in a higher sensitivity to different stresses, such as oxidative stress. Upon the addition of H_2_O_2_, Msn2/4 are transported to the nucleus [56]. Msn2/4 are required for activation of several genes such as *CTT1* (coding for cytosolic catalase) through a stress response element (STRE) consisting of a pentameric core of CCCCT. Msn2/4 sensing of H_2_O_2_ is accomplished through PKA-dependent phosphorylation [11].

Maf1 is a transcriptional repressor of RNA polymerase III (Pol III) conserved from yeast to mammals. Yeast cytosolic thioredoxins 1 and 2, Trx1 and Trx2, are essential to trigger the nuclear accumulation of Msn2/4 and Maf1 specifically under H_2_O_2_ treatment. However, Maf1 dephosphorylation and its subsequent nuclear accumulation in response to H_2_O_2_ do not seem to occur via repression of PKA-dependent phosphorylation, as for Msn2, but instead seem to be regulated by the PP2A phosphatase [56].

It has been shown that antioxidant proteins may act as a transcription factors under specific conditions. Notably, the cytosolic Sod1 superoxide dismutase is well known not only for its anti-oxidant activity, but also because of its involvement in amyotrophic lateral sclerosis, in cancer and in other diseases. In response to H_2_O_2_, Sod1 is transported to the nucleus in yeast and in humans where it binds to promoters and regulates a program of gene expression encompassing, e.g., the genes *RNR3,* encoding a subunit of ribonucleotide reductase, and *GRE2*, encoding a stress-related aldehyde reductase, among others [57]. In mammals, the transcriptional activity of *STAT3* (a transcription factor signal transducer and activator of transcription) is decreased upon H_2_O_2_ addition through a disulfide bond with peroxiredoxin 2, Prx2 [58]. 

Another way that transcription is modulated upon H_2_O_2_ addition targets RP. RP synthesis is a major consumer of cellular resources, and its regulation is of utmost importance. yeast split finger protein, Sfp1, is a transcription factor that regulates the transcription of ribosomal protein and biogenesis genes. Under normal conditions, Sfp1 accumulates in the nucleus and promotes RP gene expression. Upon inhibition of the TOR pathway, stress or changes in nutrient availability, Sfp1 leaves the nucleus and RP gene transcription is inhibited. Rosa M et al. showed in 2004 that Sfp1 responds to different stress conditions, including the addition of H_2_O_2_, during which Sfp1 is localized in the cytosol and RP gene expression is inhibited. The authors suggested that Sfp1 localization responds to nutrient availability dependent on the TOR and PKA pathways [59].

## 5. Roles of H_2_O_2_ in Protein Synthesis: Translation

Although regulation of translation can happen at any level of protein synthesis, most regulatory events are thought to occur at the initiation step. The initiation steps require the coordinated action of multiple factors, many of which are targets of tight regulation [38]. Translation initiation is a complex process involving at least 12 eukaryotic initiation factors (eIFs) interacting with ribosomal subunits, initiator methionyl-tRNA (Met-tRNA_i_
^Met^) and mRNAs [42].

The translation pathway in cells can be divided into four stages: initiation, elongation, termination and ribosomal recycling. In eukaryotes, the main eukaryotic cellular translation components are the two ribosomal subunits, 40S and 60S, composed of four RNAs and over 80 proteins, such as mRNAs and aminoacyl-tRNAs (aa-tRNAs), formed by attaching each of the 20 amino acids to its specific tRNA through an energy-consuming reaction catalyzed by specific aminoacyl-tRNAs synthetases; protein factors that interact with the other components and GTP and ATP, providing energy through their hydrolysis. Most of the principal features of the protein synthesis pathway are common to all domains of life, especially the structure of the ribosome responsible for aa-tRNA binding to mRNA codons and peptide bond formation [37].

### 5.1. Translation Initiation

eIF2α, a guanine nucleotide-binding factor binding to a GTP, interacts with the initiator methionyl-tRNA (Met-tRNA_i_^Met^) to form a ternary complex (eIF2–GTP–Met-tRNA_i_^Met^) that is competent for translation initiation. eIF2 is released from the ribosome as a binary complex with GDP. GDP is replaced by GTP in a guanine nucleotide exchange reaction promoted by eIF2B. In yeast, phosphorylation of eIF2α by the protein kinase Gcn2 converts eIF2 from a substrate to an inhibitor of the guanine nucleotide exchange factor eIF2B. This results in decreased levels of both eIF2B and ternary complex. In response to low levels of ternary complex, the translation of Gcn4 is activated, a mechanism involving its short upstream inhibitory open reading frames [60]. Gcn4 is a transcription factor that activates the gene expression of many targets, including amino acid biosynthetic genes, and inhibits the expression of RP genes [61]. Upon Gcn4 activation, transcriptional reprogramming for the adaptation to nutrient starvation occurs.

In the regulation of translation initiation, there are three main stress-signaling pathways: the integrated stress response (ISR) with the eukaryotic translation initiation factor 2, eIF2, the TOR pathway acting through eIF4E and the mitochondrial stress response pathways, including the mitochondrial unfolded protein response (UPR^mt^). Translation of most eukaryotic mRNAs responds to the action of both ISR and mTOR, such that interference with both of these pathways affects most of the translatome.

#### 5.1.1. The Integrated Stress Response

In mammalian cells, there are four protein kinases that inhibit translation initiation by phosphorylating eukaryotic initiation factor 2 (eIF2), GCN2 (general control nonderepressible kinase), PKR (double-stranded protein kinase activated by RNA), HRI (heme-regulate inhibitor) and PERK/PEK (PKR-like endoplasmic reticulum eIF2 kinase), and they are regulated under different stress conditions [62]. In yeast, Gcn2 is the only kinase phosphorylating eIF2α, and this occurs at a conserved serine (Ser 51) residue in response to nutrient starvation and other stresses (Figure 2).

Upon Gcn4 activation, transcriptional reprogramming for the adaptation to nutrient starvation occurs. Many of the proteins that are activated upon increased levels of phosphorylated eIF2α (eIF2α-P) are key ISR regulators (*ATF4* for example) that affect diverse cellular processes similar to the transcription factor Gcn4 in yeast. In addition, many of these mRNAs of ISR genes feature upstream ORFs, which inhibit translation of the downstream ORFs including the primary coding sequence (CDS). Consequently, when the levels of eIF2α-P are low and ternary complex (TC) abundant, ribosomes initiate at 5′ proximal ORFs and then terminate before reaching the CDS. In contrast, when the levels of eIF2α-P are high and the levels of the TC are low, reinitiation at ORFs becomes less frequent, which allows the scanning 40S ribosomal subunit to reach a CDS and eventually translate *GCN4* (for more information we refer the reader to the excellent review by Advani et al. [38]) (Figure 2).

In yeast, oxidative stress caused by exposure to H_2_O_2_ results in a rapid and reversible inhibition of protein synthesis mediated in part by Gcn2-dependent phosphorylation of eIF2α. This stress condition has been proposed to activate Gcn2 through a variety of mechanisms: oxidation of amino acids that cause amino acid starvation, oxidation of proteins and nucleic acids necessary for tRNA aminoacylation, oxidation of tRNAs and more [42].

There is a shift of ribosomes from polysome fractions into monosomes following treatment with H_2_O_2_ addition, which means that the initiation of translation is reduced. In addition, decreased ribosomal runoff has been observed following H_2_O_2_ addition, consistent with an inhibition of translation elongation and/or termination [62]. H_2_O_2_ was also shown to cause an upregulation of *GCN4* expression, at least in the divergent W303 yeast strain as well as a requirement for Gcn4 for H_2_O_2_ tolerance [64]. On the other hand, Gcn4 is neither activated by H_2_O_2_ nor required for H_2_O_2_ resistance in two other yeast strain backgrounds (S288c and CEN-PK) [65,66], suggesting that other Gcn2 targets may be more important in the ability of cells to adapt to H_2_O_2_ in these strains. Interestingly, we recently showed that cells engineered to secrete increased levels of recombinant protein displayed increased levels of H_2_O_2_ in the cytosol and Gcn2 activation [66]. Notably, Gcn2 deficiency restored H_2_O_2_ levels, cytosolic protein synthesis and ER-folding homeostasis through an unknown mechanism, suggesting a role for H_2_O_2_ and Gcn2 in balancing cytosolic protein synthesis to protein secretion in the ER.

Using ribosome profiling coupled with next-generation sequencing, Gerashchenko et al. studied the interplay between transcription and translation upon H_2_O_2_ treatment [67]. H_2_O_2_ led to a massive and rapid increase in ribosome occupancy of short upstream ORFs and of the N-terminal regions of ORFs that preceded the transcriptional response. The authors identified a different transcription response after 5 or 30 min of H_2_O_2_ treatment and significant differences between mRNA abundance and its translation. In fact, some mRNAs were not translated at all. For example, *SRX1* (coding for sulfiredoxin) is present in unstressed yeast cells as a moderately transcribed gene with no detectable ribosomal occupancy. Its transcription increases immediately after the addition of peroxide [67]. In fact, we showed already in 2011 that *SRX1* is translationally regulated upon the addition of H_2_O_2_ and that its translation is more efficient in cells grown under caloric restriction (or cells expressing low PKA activity) [65]. Srx1 is required to reduce/reactivate hyperoxidized Tsa1 and to extend lifespan. In mammalian cells, Yin Young Baek et al. showed that Srx has a role in maintaining the balance between H_2_O_2_ production and elimination and in protecting cells from apoptosis induced by low steady-state levels of H_2_O_2_. Deletion of Srx in A549 human lung cancer cells and wild-type mouse embryonic fibroblast (MEF) cells elicited a dramatic increase in intra- and extracellular H_2_O_2_, sulfinic acid 2-Cys Prxs and apoptosis [68].

#### 5.1.2. The TOR Pathway

Other key points of regulation in translation initiation are via regulated binding of initiation factors to the mRNA cap. The TOR pathway is the second major regulator contributing to translational control under stress. Under normal conditions, the initiation factor eIF4E recognizes and binds to the 5′ cap structure of mRNAs and, together with eIF4G and eIF4A, forms the eIF4F complex. Then, eIF4G interacts with the poly(A)-binding protein (PABP) deposited at the mRNA 3′end. eIF4G also recruits eIF4A to the cap for local unwinding of mRNA secondary structure elements (Figure 3). Unwinding of mRNA and interactions of eIF4G with eIF3, eIF5 or eIF1 facilitate the recruitment of the 43S preinitiation complex. Activation of the mTOR pathway leads to phosphorylation of eIF4E-binding protein (4E-BP), which competes with eIF4G for binding to eIF4E. As a result, phosphorylation of 4E-BP leads to its release from eIF4E and consequently stimulates cap-dependent translation initiation [69,70,71].

In mammalian cells, H_2_O_2_ and oxygen deprivation (hypoxia) inhibit protein synthesis through the inhibition of 4E-BP [72].

A recently published paper by Klann K showed, using multiplexed enhanced protein dynamics (mePROD) proteomics, a global rearrangement of cellular translation upon various treatments modulating eIF2α and/or mTORC1 activity, and that the ISR and the mTORC1 pathways regulate translation of a partially overlapping set of proteins, despite their distinct upstream regulation in different stress conditions.

#### 5.1.3. Mitochondrial Stress Response Pathways

As we noticed previously in this review, the most important source of H_2_O_2_ is the electron transport chain in mitochondria. Recent studies show that mitochondrion-derived ROS can modulate cytosolic protein synthesis through oxidative modification of ribosomal proteins [41]. Furthermore, the mitochondrial unfolded protein response (UPR^mt^) has been described in flies, worms and mammals [73]. When UPR^mt^ is activated (for example, during impairment of mitochondrial ribosomes or upon high levels of ROS) there is a decrease in mitochondrial membrane potential and a decreased efficiency of mitochondrial protein import, which activates the proteasome and inhibits cytosolic translation [74]. This activation has not been described yet in the yeast *S. cerevisiae*. Mitochondrial stress activates the ISR pathway, and there are changes in the transcriptome, proteome and proteasome activation, all eliciting beneficial effects in helping cells adapt to these conditions [41,74].

The cytosolic translational responses depend on the duration of mtUPR signaling. Upon short-term mitochondrial stress, protein synthesis is inhibited in a reversible way because of the reversible oxidation of ribosomal proteins and inhibition of mTOR (via a decrease in the phosphorylation of S6K1 and 4E-BP1 proteins) in an eIF2α-phosphorylation independent manner. In contrast, under conditions of long-term mitochondrial stress (by long-term chemical stress or knockdown of oxidoreductases), protein synthesis appears to be inhibited in an eIF2α-dependent, but likely ROS-independent manner [75].

### 5.2. Translation Elongation

Ribosomes contain three tRNA-binding sites, designated the aminoacyl (A) site, where the aminoacyl-tRNA binds, the peptidyl (P) site, where the peptide bond is formed, and the exit (E) site, where the deacylated tRNA exits the ribosome. First, the initiator tRNA binds to the P site of the small ribosomal subunit and the A site is poised to receive an aminoacyl-tRNA. In the elongation phase each additional aminoacyl-tRNA brings each amino acid to the A site of the ribosome, where it is added to the nascent polypeptide chain through a peptide bond in the P site leading the mRNA to advance by one codon. To facilitate this process, elongation factors 1 (eEF1A and eEF1B) and elongation factor 2 (eEF2) aid.

Attenuating translational elongation in response to stress conditions, as opposed to translational initiation, offers the advantage that ribosomes remain bound to mRNAs and can rapidly resume protein synthesis once stress is removed or detoxified.

As we have noted previously, responses to low and high levels of H_2_O_2_ are different. In yeast, under low concentrations of H_2_O_2_ (0.2mM) there is an adaptative response with an increase in the synthesis of key proteins, such as antioxidants (Ctt1, Gpx2, Trr1, Srx1 and others). However, upon the addition of higher levels of H_2_O_2_ (from 0.8 mM) there is an increase, around 50%, in ribosome association with mRNAs encoding ribosome biogenesis and rRNA processing (among other) components [63], mRNAs which can subsequently rapidly resume translation upon restored oxidative homeostasis. The ribosomes remain bound to mRNA, and a decrease in ribosomal runoff observed under these conditions is consistent with an inhibition of translation, elongation or termination [63]. Moreover, ribosomal proteins can act as sensors for H_2_O_2_, mediating a decrease in protein synthesis by pausing the translation at the post-initiation stage. A depletion of specific ribosomal proteins partially prevented translation attenuation, and ribosomal protein cysteine oxidation may be reversed once favorable conditions reappear in order to restore protein synthesis [41].

Regulation of elongation could occur at the level of eEF1A and B upon H_2_O_2_ exposure. For example, in yeast Tef2 and Tef5 (eEF1α and eEF1β), initiation factor Nip1 (a subunit of initiation factor eIF3) and the small ribosomal subunit protein Rps5 have been noted to be modified by protein S-thiolation in response to high levels of H_2_O_2_ (2 mM) [64] (Figure 4).

Besides regulating translation initiation, the mTOR pathway also controls translational elongation through activation of elongation factor 2 (eEF2). In eukaryotic cells, eEF2 promotes the GTP-dependent translocation of the ribosome along the mRNA. eEF2 undergoes phosphorylation at Thr56 by a highly specific eEF2 protein kinase, which inactivates eEF2. Activation of mTOR promotes phosphorylation of the eEF2 kinase at least at three sites (Ser366, Ser359 and Ser78), which leads to its inactivation. As a consequence, mTOR activates eEF2 and thereby enhances translation elongation [76]. Other well-known targets of TOR signaling are the S6 kinases, S6K1 and S6K2. Activation of TOR leads to activation of both the kinases S6K1 and S6K2, which subsequently phosphorylate several components of the translation machinery, including the 40S ribosomal proteins S6, eIF4B and the eEF2 kinase [77].

Another point of regulation in the elongation phase is at the level of tRNAs. tRNAs are the second most abundant noncoding RNA and represent around 10–15% of total RNA in human cells. tRNAs are expensive to synthesize and their expression is tightly regulated [38]. Regulating the abundance of specific tRNA pools presents a dynamic mechanism of modulating translation. One mechanism by which the regulation of tRNA transcription is achieved is through *MAF1*, that as we noticed before is a constitutive repressor of polymerase III transcription (Pol III), regulated by TORC1 and PKA in yeast. Starvation inhibits mTOR, inducing a dephosphorylation of Maf1 which translocates into the nucleus and inhibits recruitment of Pol III on tRNA genes. This causes decreased tRNA transcription and an increase in the transcription of housekeeping stable tRNAs that alters the cellular tRNA pool to favor translation of stress-responsive proteins [38]. Interestingly, Pol III has been suggested to regulate the lifespan of flies downstream of TORC1, supporting a key role in the aging process [54].

A paper published in 2021 using ribosome profiling showed in the yeast *Schizosaccharomyces pombe* that ribosomes stall on tryptophan codons upon the addition of H_2_O_2_ via an unknown mechanism, and that this affects translation elongation [78]. Moreover, specific modifications in the tRNA anticodon loop are vital to reprogram translation under stress [79]. Modification of the wobble base in the anti-codon greatly alters the kinetics of the codon:anticodon interactions, thereby altering codon-dependent stability and translation of specific transcripts. Oxidative stress has multiple effects on tRNA metabolism, including regulating methylation at the wobble position (m^5^UC), tRNA misacylation and tRNA cleavage. The most common modification of tRNA at the wobble position is methylation of nucleotide bases. Cells modulate methylation and demethylation of tRNA bases to tune translational accuracy in response to stress. For example, in *S cerevisiae*, Trm9-mediated 5-methyl-U (m^5^U) modifications at positions U34 of tRNA ^Glu (UUC)^ and tRNA ^Arg (UCU)^ are important for maintaining translational fidelity in response to DNA damage [80]. Another wobble base modification, 5-methyl-C (m^5^c) at position C34 of tRNA^Leu (CAA)^ catalyzed by methyltransferase Trm4, contributes to codon-based translation of stress response proteins under H_2_O_2_ conditions [81]. Thus, the increase in methylation at position C34 of tRNA^Leu (CAA)^, in response to H_2_O_2_, enhances the efficiency of translation of mRNAs enriched in the UUG codon recognized by this tRNA (as *RPL22A* and *RPL22B*) and causes a change in the ribosome composition and selective translation of genes enriched with the codon TTG. tRNA wobble uridines are chemically modified at position 2 in the wobble uridine with a thiol (s^2^) and commonly at position 5 with a methoxycarbonylmethyl (5-methoxycarbonylmethyl-, mcm^5^-) modification, and these modifications are usually present together. Modifying tRNAs in this way regulates overall carbon and nitrogen metabolism dependent on methionine availability [82]. Mutants deficient in these modifications show relatively minor effects on translation but still exhibit clear phenotypes, including higher sensitivity to oxidative stress. The components of the Urm1 pathway in *S. cerevisiae* catalyze thiolation of uridine bases (mcm^5^s^2^U) at the wobble position 34 of tRNA ^lys(UUU)^, tRNA ^Glu(UUC)^ and tRNA ^Gln(UUG)^. These modifications promote efficient translation of transcripts enriched in AAA, GAA and CAA codons. Laxman et al., 2013, showed that the thiolation status of the wobble uridine (U_34_) nucleotides (present in lysine, glutamine and glutamate tRNAs) is modified as a function of methionine and cysteine availability and that this subtle modification of these specific tRNAs regulates cellular translational capacity and metabolic homeostasis [82]. ROS furthermore induce methionine misacylation in HeLa cells, where approximately 1% of the cellular tRNAs were methionine misacylated. This results in increased incorporation of methionine in protein, a heterogeneity that has been shown to promote cell viability [82].

### 5.3. Translation Termination

Translation termination must proceed with a high degree of fidelity to avoid errors in the decoding process and this is important since the suppressed reading of stop codons (readthrough) may have deleterious effects on the proteome. Translation termination occurs when a stop codon (UAA, UAG and UGA) is translocated into the A site of the ribosome (Figure 5).

There are two important eukaryotic release factors, eRF3 and eRF1. There is a regulation of the translation termination at the level of the eRF3. eRF3 (or Sup35 in yeast) is also uniquely able to form prion aggregates known as *[PSI^+^]*. Due to different genetic and environmental factors, as well as upon the addition of H_2_O_2_, *[PSI^+^]* aggregates increase. Under these conditions, eRF3 can no longer perform its normal function in translation termination and there is elevated readthrough of termination codons [83]. Sideri et al. showed in 2010 that the double mutant of the yeast peroxiredoxins Tsa1 and Tsa2 suffers an increase in de novo formation of *[PSI^+^]* and that levels are increased even further upon the addition of H_2_O_2_ [83].

### 5.4. Ribosomal Recycling

Translation finishes when the nascent polypeptide is released from the ribosome during the termination step, but the ribosomes still remain bound to the mRNA. Ribosomes are ultimately released from the mRNA and split into subunits that can bind mRNA anew, a process that is called ribosome recycling. This step is essential for the viability of cells. Recycling mechanisms differ among organisms in the three domains of life [84,85]. For example, in eukaryotes and archaea, termination is carried out by the release factor and a translational GTPase (eRF1 and eRF3). eRF1 remains in the ribosome after the release of the nascent peptide and helps to recruit the factor that catalyzes subunit splitting (Rli1 in yeast or ABCE1 in mammals) [86]. Rli1 is an iron–sulfur (FeS) cluster domain-containing 80S recycling factor that is essential for eukaryotic cell viability and has a role in ribosomal recycling, among other functions in translation (e.g., ribosome biogenesis and translation initiation). Rli1/ABCE1 is inactivated under H_2_O_2_ conditions through its FeS domain [87], thus being able to sense H_2_O_2_, a function that may contribute to the inhibition of protein synthesis under H_2_O_2_ conditions. The use of ribosome profiling in *S. cerevisiae* revealed that depletion of Rli1 results in the accumulation of ribosomes at the stop codons and in the 3′UTR [88].

## 6. Conclusions

All in all, hydrogen peroxide impacts in various ways on all stages of protein synthesis and is well situated, at low endogenously produced levels, to coordinate intracellular functions across compartment boundaries. In this respect, it is relevant to look a bit further into the properties and roles of second messengers, molecules and ions that quickly relay signals received from extracellular or intracellular stimuli to effector proteins. Second messengers have diverse properties that allow signaling within membranes (hydrophobic molecules), within compartments, e.g., the cytosol (polar molecules) or both as is the case of gases. The intracellular levels of second messengers are tightly regulated, and in unstimulated cells they are generally found in low concentrations. However, upon stimulation, concentrations typically swiftly rise, allowing the second messengers to rapidly diffuse from their source to the ligand-specific protein sensors [89]. In this context, H_2_O_2_ could be argued to perform a second messenger role in coordinating protein synthesis with, e.g., mitochondrial activity or ER-folding capacity [41,66]. Signal attenuation upon repeated stimulation (adaptation) is an inherent property of signaling systems involving adaptive changes in the equilibrium state of biological systems in response to alterations in their surrounding environment [90]. In this respect, adaptive responses to H_2_O_2_ involving transcription and more long-term adjustments of H_2_O_2_ homeostasis could be argued to perform roles similar to negative feedback loops, ensuring signal attenuation upon repeated stimulation [89].

## Figures and Tables

**Figure 1 antioxidants-10-00952-f001:**
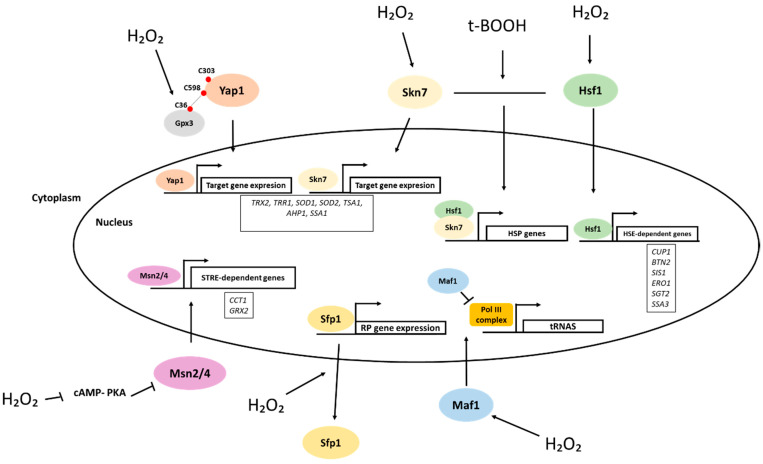
Impact of H_2_O_2_ on transcription in yeast. Five different transcription factors respond to H_2_O_2_ in yeast. Upon H_2_O_2_ addition, Yap1 is retained in the nucleus where it stimulates the expression of target genes. Gpx3 is the H_2_O_2_ sensor of Yap1 that uses its cysteine 36 to form a disulfide bond with the cysteine C598 of Yap1, eventually resolved into a Yap1 intramolecular disulfide bond (between C598 and C303) inhibiting nuclear export. Yap1 controls the expression of different genes most of which require the presence of Skn7 as well. Upon the addition of t-butyl hydrogen peroxide (t-BOOH), Hsf1 interacts with Skn7 to achieve maximal induction of several HSP genes. Msn2/4 sensing of H_2_O_2_ is accomplished through PKA-dependent phosphorylation. Maf1 controls the availability of tRNAs under H_2_O_2_ through binding to PolIII. Under H_2_O_2_ conditions, Sfp1 is localized in the cytosol and RP gene expression is inhibited.

**Figure 2 antioxidants-10-00952-f002:**
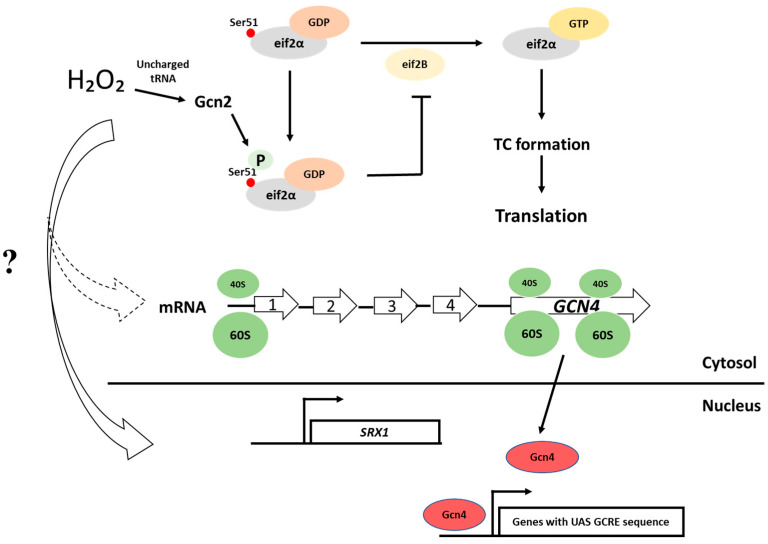
Impact of H_2_O_2_ on translation initiation. Integrative stress response. Under H_2_O_2_, eIf2α is phosphorylated by Gcn2, ternary complex levels decrease and translation is inhibited. Moreover, the translation of *GCN4*, a transcription factor that activates gene expression in response to stress, may be activated, at least in the divergent w303 yeast strain background [63]. In addition, upon Gcn2 activation, sulfiredoxin, Srx1, is more efficiently translated upon addition of H_2_O_2_ to cells grown under caloric restriction [63].

**Figure 3 antioxidants-10-00952-f003:**
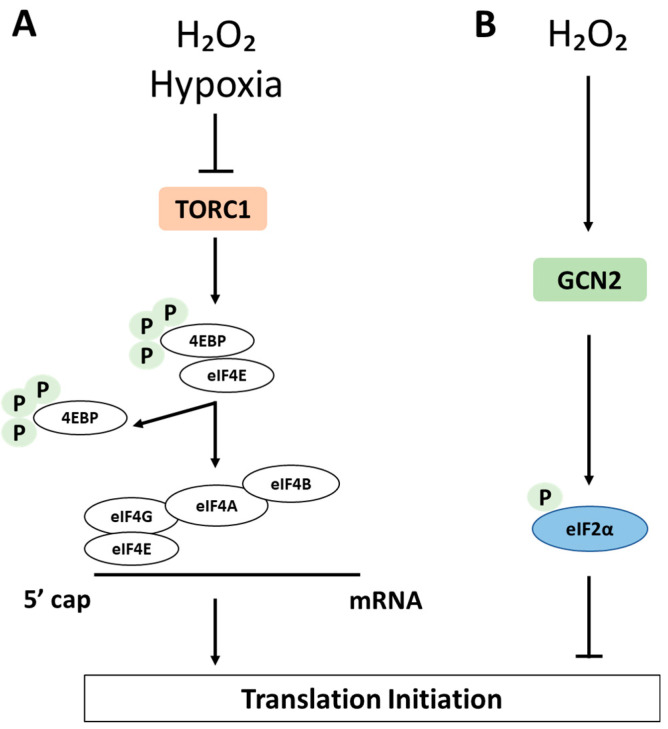
Impact of H_2_O_2_ function on translation initiation. The TOR pathway. In mammalian cells, H_2_O_2_ and hypoxia inhibit protein synthesis through inhibition of 4E-BP (**A**). There is a partial overlap in target genes of both pathways for translation inhibition under H_2_O_2_ conditions (**B**).

**Figure 4 antioxidants-10-00952-f004:**
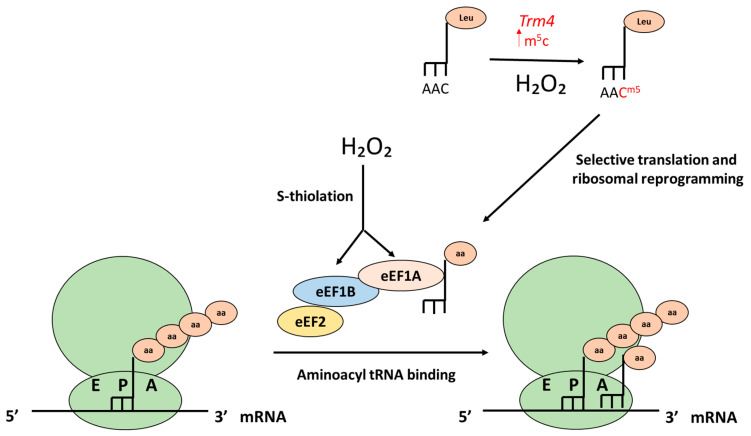
Effects of H_2_O_2_ on translation elongation. In yeast, eIE1A and eIE1B can be modified by protein S-thiolation in response to high concentrations of H_2_O_2_. Increase in the methylation in the wobble base 5-methyl-C (m^5^c) at position C34 of tRNA ^LEU(CAA)^ in response to H_2_O_2_ enhances the efficiency of translation of genes enriched in the UUG codon (as *RPL22A* and *RPL22B*) and causes a change in ribosome composition.

**Figure 5 antioxidants-10-00952-f005:**
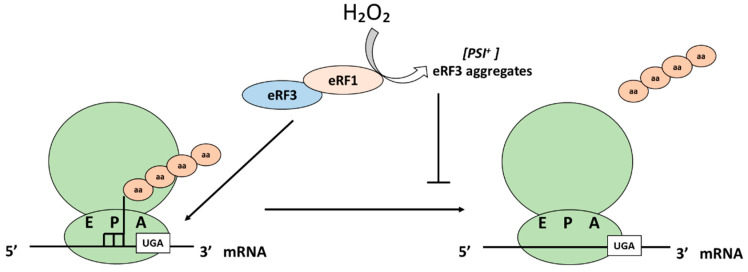
Effect of H_2_O_2_ on translation termination. Eukaryotic release factor 3, eRF3, forms prion aggregates known as *[PSI^+^]* under H_2_O_2_ conditions and can no longer perform its normal function in translation termination, leading to an elevated readthrough of termination codons.

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
