# Peer review of "Impact of Hydrogen Peroxide on Protein Synthesis in Yeast"

_antioxidants, 2021, doi:10.3390/antiox10060952_

Round 1

Reviewer 1 Report

Major criticisms:

  1. The paragraph “Systems involved in responses to H2O2” is confused and not easy to read. Perhaps, the division in sub-paragraphs could help the reader.
  2. Most of the information along the manuscript regards yeast. I suggest to change the title in “Impact of H2O2 on protein synthesis in yeast”
  3. Authors cannot assert “the role of GSH as an antioxidant is unclear” (line 110), dismissing in a too small paragraph the most important redox regulator in cells (reporting also a dated literature). The only recognized function for GSH concerns its involvement in Fe-S biogenesis, but the GSH functions, as direct ROS scavenger, enzymes cofactor/substrate, protein regulator and modulator of redox signalling, are well established (just as an example, see Muri J, Kopf M. Redox regulation of immunometabolism. Nat Rev Immunol. 2020 Dec 18. doi: 10.1038/s41577-020-00478-8. Epub ahead of print. PMID: 33340021).  
  4. The sentence “a common response to stress conditions is the global inhibition of protein synthesis” (line 163) is not correct. There are many stressed conditions (for instance, in cancer or in redox imbalance), where some proteins are over-expressed.
  5. The paragraph 3 on “Protein synthesis regulation” does not offer a wide view on this important issue that, given the title of the review, I would suggest to focus on “Redox regulation of protein synthesis”.
  6. The paragraph 4 on the” role of H2O2 in transcription” totally ignores NRF2, the master regulator of antioxidant response in cells.
  7. Overall, it is not clear the role of H2O2 on protein synthesis: does it work as a second messenger? Is it good or bad? Does its role depend on cell conditions? And what happens under pathological conditions?
  8. The manuscript appears as a long list of transcription factors and proteins, not particularly informative.
  9. Finally, but not negligible, the manuscript is heavy to read and needs a substantial revision of the English Language.

Author Response

  1. The paragraph “Systems involved in responses to H2O2” is confused and not easy to read. Perhaps, the division in sub-paragraphs could help the reader.

We thank the reviewer for this comment and have clarified the text e.g. by more properly defining its subdivision into cysteine thiol-redox and H2O2 detoxification systems relevant to H2O2 homeostasis. In addition, we have introduced these two subdivisions more clearly in the text through the use of bold font (lines 96-117).

2. Most of the information along the manuscript regards yeast. I suggest to change the title in “Impact of H2O2 on protein synthesis in yeast”

We have changed the title as per the reviewer’s suggestion.

3. Authors cannot assert “the role of GSH as an antioxidant is unclear” (line 110), dismissing in a too small paragraph the most important redox regulator in cells (reporting also a dated literature). The only recognized function for GSH concerns its involvement in Fe-S biogenesis, but the GSH functions, as direct ROS scavenger, enzymes cofactor/substrate, protein regulator and modulator of redox signalling, are well established (just as an example, see Muri J, Kopf M. Redox regulation of immunometabolism. Nat Rev Immunol. 2020 Dec 18. doi: 10.1038/s41577-020-00478-8. Epub ahead of print. PMID: 33340021).  

As we mentioned previously, at least in yeast (not in other organisms as further specified in the revised text), the roles of GSH either as an antioxidant or as a redox regulator are unclear, because it was shown that neither cells depleted of or containing elevated, toxic levels of GSH, were affected in thiol-redox maintenance except for higher levels of GSH blocking oxidative protein folding and secretion. This is consistent with the inability of yeast to synthesize selenocysteine. We have clarified this further in the text (lines 112-113 and 121-124) and point the peculiarity of yeast in this respect out more clearly in the revised version.

4. The sentence “a common response to stress conditions is the global inhibition of protein synthesis” (line 163) is not correct. There are many stressed conditions (for instance, in cancer or in redox imbalance), where some proteins are over-expressed.

The idea that stress elicits a global inhibition of protein synthesis is not new and shared by many of the cited papers (eg. [32] and [33]). We discuss to some extent the remodeling and reprogramming of protein synthesis upon H2O2 (lines 188-199) a feature that is perfectly compatible with some proteins being overexpressed upon stress in cancer and/or redox imbalance. Thus, the statement is not to be read as that the synthesis of each and every protein is reduced but rather that 1) the activity of the protein synthesis machinery is lower following stress and 2) most proteins are synthesized to lower levels. Protein synthesis is still active following stress to support synthesis of some proteins required to boost cell survival and growth. For example, in translation initiation, under stress conditions, there is an inhibition of global translation but some genes (as GCN4) are only translated under this condition (Figure 2).

5. The paragraph 3 on “Protein synthesis regulation” does not offer a wide view on this important issue that, given the title of the review, I would suggest to focus on “Redox regulation of protein synthesis”.

We corrected the title of this paragraph according to the suggestion from the reviewer.

6. The paragraph 4 on the” role of H2O2 in transcription” totally ignores NRF2, the master regulator of antioxidant response in cells.

We thank the reviewer for this comment. We added short information about transcription factors responding to H2O2 in higher eukaryotes and about the role of NRF2 in particular (lines 223-229).

7. Overall, it is not clear the role of H2O2 on protein synthesis: does it work as a second messenger? Is it good or bad? Does its role depend on cell conditions? And what happens under pathological conditions?

We stated in lines 620-625 that low levels of H2O2 fulfil two key roles of second messengers. H2O2 is a regulator of expression of a lot of genes through its action of TFs under different conditions. That means that it can act as a messenger to modulate gene expression in cell adaptation and survival. This molecule can also more directly modulate gene translation. In the revised manuscript we also directly highlight two examples of endogenously produced H2O2 being involved in modulating translation via the here discussed pathways (lines 199-208 and 411-416, respectively).

8. The manuscript appears as a long list of transcription factors and proteins, not particularly informative.

In this review we focus for the first time on an overview of the role of H2O2 in transcription and translation in the protein synthesis process in yeast. A few reviews available to date focus on the role of H2O2 in translation but we would here like to show in one document that there is a role of H2O2 in both transcription and translation that are involved in different aspects of the response to H2O2. Furthermore, because the concept of H2O2 modulating protein synthesis is rather new we believe that it is premature to call out for excessive discussions of molecular interconnections at this stage and hope studies in the near future will bolster with further molecular details of this kind what may here, at least for this reviewer, appear as a long list of components.

9. Finally, but not negligible, the manuscript is heavy to read and needs a substantial revision of the English Language.

We thank the reviewer for noticing. We have had the text read through and edited appropriately by a fluent user of the English language.

Reviewer 2 Report

The manuscript shows the effect of cellular stress condition on different phases of protein synthesis. The work is interesting and well structured. The reported schemes are well reported and help the reader in the complexity of the topic. 

Author Response

We thank the reviewer for the comments

Reviewer 3 Report

The objective of this review is of relevance, that is, updating the existing studies on the impact of oxidative stress (extensive to other stresses) on the machinery of protein biosynthesis and the signalling pathways involved. This objective also includes the previous description of the oxygen species implicated in oxidative stress, with emphasis on hydrogen peroxide, and the chemical damage caused on protein residues, as well as the protective enzymatic systems in the cell. Globally, a review like this is valuable and deserves publication. However, in its present form it contains significant defects, especially grammatical ones, which sometimes make difficult understanding what authors mean in some sentences or paragraphs. It therefore requires an extensive revision before publication. Below I distinguish between comments on the scientific contents (a few) and on grammatical and style errors (many).

Comments on the contents:

-Page 3, first paragraph. It is not exact that the glutaredoxin system is directly involved in H2O2 homeostasis. While this may be the case for the thioredoxin system through the involvement as hydrogen donor for peroxiredoxins, glutaredoxins are mainly involved (as the authors correctly point) in controlling the redox state of protein thiol groups. Different is the case for glutathione, which may be implicated in the detoxification of peroxides through the activity of glutathione-dependent peroxidases.

-Page 3. For clarification to non-experts, the relationship between thioredoxin peroxidases and peroxiredoxins should be indicated.

-Fig. 1 legend and the corresponding text in line 231. The Cys residues involved in Yap1 intramolecular disulfide bond formation could be indicated.

-Paragraph from line 314 to 319 concerns the initiation stage of translation. Therefore, it should be moved to the beginning of section 5.1.

Other comments, mainly on style and grammatical errors (non-exhaustive list):

-Line 13: …reactive oxygen species (ROS)…

-Line 13: …but also at other…

-Line 23: electron acceptor

-Line 31: …the name of ROS refers to…

-Line 49 to 53: Please rewrite this complex sentence

-Line 68: reactive species

-Line 73: secondary messenger

-Line 79: there are…

-Line 80: signaling molecule…

-Line 93: ROS detoxification systems

-Line 95: different levels of protein synthesis

-Line 116: Write ‘Prx’ as an abbreviation into parenthesis the first time the term ‘peroxiredoxin’ is employed, and then be consistent about employing the abbreviation or the full name

-Line 119: Be consistent about using 1-Cys or 1-cys, and 2-Cys or 2-cys (always the same style!)

-Line 170: multisubunit complexes

-Line 174-175: …¿promoting gene transcription, transcription of ribosomal proteins (RP)…?

-Lin 176: Check the sentence, since growth of the cells is not repressed upon high nutrient availability by the TORC1 pathway (unless I did not understand the sense of the sentence!)

-Line 191: …exposing yeast to…

-Line 196: …or exogenous H2O2 revealed…

-Line 207: …regulation of the activity of some transcription factors (TF)

-Line 238 and 239: Some confusion between gene nomenclature and protein nomenclature

-Line 248: a pentameric heat shock element

-Line 256: DNA binding domain

-Line 257: …causes binding of Hsf1 (it is the protein which binds, not the gene!!)

-Line 258: This is the first time in the text that this species is names as such. Therefore, write it in full: Saccharomyces cerevisiae, and then abbreviate when cited later

-Line 305: In eukaryotes, the main cellular…

-Line 307: mRNAs

-Line 338: responds to the action…

-Line 350: omit ‘factors’

-Line 366: …to cells grown under…

-Lines 357-358: …and eventually initiate GCN4 translation…

-Line 371: tRNAs

-Line 371: …that cause amino acid starvation…

-Line 373: delete ‘)’

-Line 373: …oxidation of tRNAs…

-Line 375: …following H2O2 addition…

-Line 378: W303

-Line 416: H2O2 and hypoxia inhibit protein synthesis…

-Line 427: …the most important source of H2O2 is the…

-Line 435: S. cerevisiae

-Line 440 onwards: Rewrite the sentence since a verb seems to be lacking

-Line 461: delete ‘response’

-Line 462: delete ‘protein’

-Line 464. …ribosome association to…

-Lines 471-472: Rewrite the sentence

-Line 475: ‘Rps5’ not in parenthesis

-Line 480: Increase of the methylation…

-Line 482: …enriched in the…

-Line 488: …, which inactivates eEF2.

-Lines 488-489: …at last at three sites.

-Line 504: …and an increase of the transcription…

-Line 515: The most common modification of tRNA…

-Line 517: S. cerevisiae

-Line 525: …enriched with…

-Line 539: ROS induce methionine…

-Line 540: …were methionine-misacylated

-Line 576: …to sense H2O2, a function that…

-Line 582: …on all stages of protein synthesis…

-Line 583-584: …to coordinate intracellular functions…

Author Response

Page 3, first paragraph. It is not exact that the glutaredoxin system is directly involved in H2O2 homeostasis. While this may be the case for the thioredoxin system through the involvement as hydrogen donor for peroxiredoxins, glutaredoxins are mainly involved (as the authors correctly point) in controlling the redox state of protein thiol groups. Different is the case for glutathione, which may be implicated in the detoxification of peroxides through the activity of glutathione-dependent peroxidases.

We thank the reviewer for noticing. We have clarified the distinction between glutaredoxin and glutathione-dependent peroxidases in the text accordingly 

-Page 3. For clarification to non-experts, the relationship between thioredoxin peroxidases and peroxiredoxins should be indicated.

We have made the distinction between TPxs and peroxiredoxins in the revised version of the text (lines 120-129).

-Fig. 1 legend and the corresponding text in line 231. The Cys residues involved in Yap1 intramolecular disulfide bond formation could be indicated.

We indicated the cysteines in Yap1 in the text as requested by the reviewer.

-Paragraph from line 314 to 319 concerns the initiation stage of translation. Therefore, it should be moved to the beginning of section 5.1.

This text paragraph covers fundamental components in protein translation, not just those relevant in translation initiation and therefore we kept it in the beginning of the ‘Roles of H2O2 in Translation’ paragraph.

Other comments, mainly on style and grammatical errors (non-exhaustive list):

We thank the reviewer for noticing. We have corrected all the below indicated grammatical and style errors and some more in the revised version of the manuscript.

-Line 13: …reactive oxygen species (ROS)…

-Line 13: …but also at other…

-Line 23: electron acceptor

-Line 31: …the name of ROS refers to…

-Line 49 to 53: Please rewrite this complex sentence

-Line 68: reactive species

-Line 73: secondary messenger

-Line 79: there are…

-Line 80: signaling molecule…

-Line 93: ROS detoxification systems

-Line 95: different levels of protein synthesis

-Line 116: Write ‘Prx’ as an abbreviation into parenthesis the first time the term ‘peroxiredoxin’ is employed, and then be consistent about employing the abbreviation or the full name

-Line 119: Be consistent about using 1-Cys or 1-cys, and 2-Cys or 2-cys (always the same style!)

-Line 170: multisubunit complexes

-Line 174-175: …¿promoting gene transcription, transcription of ribosomal proteins (RP)…?

-Lin 176: Check the sentence, since growth of the cells is not repressed upon high nutrient availability by the TORC1 pathway (unless I did not understand the sense of the sentence!)

-Line 191: …exposing yeast to…

-Line 196: …or exogenous H2O2 revealed…

-Line 207: …regulation of the activity of some transcription factors (TF)

-Line 238 and 239: Some confusion between gene nomenclature and protein nomenclature

-Line 248: a pentameric heat shock element

-Line 256: DNA binding domain

-Line 257: …causes binding of Hsf1 (it is the protein which binds, not the gene!!)

-Line 258: This is the first time in the text that this species is names as such. Therefore, write it in full: Saccharomyces cerevisiae, and then abbreviate when cited later

-Line 305: In eukaryotes, the main cellular…

-Line 307: mRNAs

-Line 338: responds to the action…

-Line 350: omit ‘factors’

-Line 366: …to cells grown under…

-Lines 357-358: …and eventually initiate GCN4 translation…

-Line 371: tRNAs

-Line 371: …that cause amino acid starvation…

-Line 373: delete ‘)’

-Line 373: …oxidation of tRNAs…

-Line 375: …following H2O2 addition…

-Line 378: W303

-Line 416: H2O2 and hypoxia inhibit protein synthesis…

-Line 427: …the most important source of H2O2 is the…

-Line 435: S. cerevisiae

-Line 440 onwards: Rewrite the sentence since a verb seems to be lacking

-Line 461: delete ‘response’

-Line 462: delete ‘protein’

-Line 464. …ribosome association to…

-Lines 471-472: Rewrite the sentence

-Line 475: ‘Rps5’ not in parenthesis

-Line 480: Increase of the methylation…

-Line 482: …enriched in the…

-Line 488: …, which inactivates eEF2.

-Lines 488-489: …at last at three sites.

-Line 504: …and an increase of the transcription…

-Line 515: The most common modification of tRNA…

-Line 517: S. cerevisiae

-Line 525: …enriched with…

-Line 539: ROS induce methionine…

-Line 540: …were methionine-misacylated

-Line 576: …to sense H2O2, a function that…

-Line 582: …on all stages of protein synthesis…

-Line 583-584: …to coordinate intracellular functions…

Round 2

Reviewer 1 Report

I appreciated the effort of the authors to improve the readability of the text.

One more suggestion:

it can be useful to expand the "conclusions" by better specifying the interesting "two key messenger roles" of H2O2.

Author Response

We thank the reviewer for this comment and we have expanded the conclusion (lines 618-636): All in all, hydrogen peroxide impacts in various ways on all stages of protein synthesis and is well situated, at low endogenously produced levels, to coordinate intracellular functions across compartment boundaries. In this respect, it is relevant to look a bit further into properties and roles of second messengers, molecules and ions that quickly relay signals received from extracellular or intracellular stimuli to effector proteins. Second messengers have diverse properties that allow signaling within membranes (hydrophobic molecules), within compartments e.g. the cytosol (polar molecules) or both as is the case of gases. The intracellular levels of second messengers are tightly regulated, and in unstimulated cells they are generally found in low concentrations. However, upon stimulation concentrations typically swiftly rise allowing the second messengers to rapidly diffuse from their source to the ligand-specific protein sensors [90]. In this context, H₂O₂ could be argued to perform a second messenger role in coordinating protein synthesis to e.g. mitochondrial activity or ER-folding capacity [41,66]. Signal attenuation upon repeated stimulation (adaptation) is an inherent property of signaling systems involving adaptive changes in the equilibrium state of biological systems in response to alterations in their surrounding environment [91]. In this respect, adaptive responses to H₂O₂ involving transcription and more long-term adjustments of H₂O₂ homeostasis could be argued to perform roles similar to negative feedback loops ensuring signal attenuation upon repeated stimulation [90]

Reviewer 3 Report

The authors have addressed correctly the questions raised by this reviewer

Author Response

We thank the reviewer for the comments